# SAD-1 kinase controls presynaptic phase separation by relieving SYD-2/Liprin-α autoinhibition

**Nathan A. McDonald[1¤a], Li Tao[1¤b], Meng-Qiu Dong[2], Kang Shen [1,3]***

**1** Department of Biology, Stanford University, Stanford, California, United States of America, **2** National Institute of Biological Sciences, Beijing, People's Republic of China, **3** Howard Hughes Medical Institute, Stanford University, Stanford, California, United States of America

¤a Current address: Georgia Institute of Technology, Atlanta, Georgia, United States of America
¤b Current address: The Hong Kong University of Science and Technology, Guangzhou, People's Republic of China

* kangshen@stanford.edu

## Abstract

Neuronal development orchestrates the formation of an enormous number of synapses that connect the nervous system. In developing presynapses, the core active zone structure has been found to assemble through liquid–liquid phase separation. Here, we find that the phase separation of *Caenorhabditis elegans* SYD-2/Liprin-α, a key active zone scaffold, is controlled by phosphorylation. We identify the SAD-1 kinase as a regulator of SYD-2 phase separation and determine presynaptic assembly is impaired in *sad-1* mutants and increased by overactivation of SAD-1. Using phosphoproteomics, we find SAD-1 phosphorylates SYD-2 on 3 sites that are critical to activate phase separation. Mechanistically, SAD-1 phosphorylation relieves a binding interaction between 2 folded domains in SYD-2 that inhibits phase separation by an intrinsically disordered region (IDR). We find synaptic cell adhesion molecules localize SAD-1 to nascent synapses upstream of active zone formation. We conclude that SAD-1 phosphorylates SYD-2 at developing synapses, activating its phase separation and active zone assembly.

**Data Availability Statement:** All relevant data are within the paper and its Supporting Information files. SAD-1 phosphoproteomics data is available at ProteomeXchange (PXD043081) and in vitro

## Introduction

Developing neurons extend polarized axons and dendrites great distances to make connections with partner cells and wire the nervous system. As partners are identified, synaptic junctions are formed that enable neuronal communication. Each synapse builds specialized pre- and postsynaptic machinery to support extremely rapid asymmetric communication across the junction [1]. In postsynapses, postsynaptic densities cluster receptors in order to receive and propagate incoming neurotransmitter signals. In presynapses, synaptic vesicles containing neurotransmitters are clustered and primed for rapid release upon incoming action potential signals [2].

The central structure of a presynapse is the "active zone," an electron-dense membrane-apposed structure marking the site of release of synaptic vesicles [2,3]. The active zone

phosphoproteomics is available at MassIVE (MSV000091471).

**Funding:** This work was funded by the Howard Hughes Medical Institute to K.S, a Helen Hay Whitney postdoctoral fellowship to N.A.M., and NIH K99 NS123233 to N.A.M. The funders had no role in study design, data collection and analysis, decision to publish, or preparation of the manuscript.

**Competing interests:** The authors have declared that no competing interests exist.

**Abbreviations:** FRAP, fluorescence recovery after photobleaching; HSN, hermaphrodite-specific neuron; IDR, intrinsically disordered region; NGM, nematode growth media; PMSF, phenylmethylsulfonyl fluoride.

comprises multiple large multivalent scaffolding proteins, including Liprin-α [4], RIM, RIM-BP, Piccolo/Bassoon, ELKS, and Munc-13 [2]. These molecules form the active zone structure and coordinate the central functions of the presynapse, including the clustering of voltage-gated calcium channels and tethering and priming of synaptic vesicles.

While the molecular composition of the active zone is well established, how this structure develops and the signaling that initiates its formation is not clear [5]. Many binding interactions have been identified that link the various active zone scaffold molecules and additional partners [2–4], suggesting their assembly into a densely bound matrix at nascent synapses. Recent evidence has additionally shown multiple active zone components are capable of liquid–liquid phase separation to form condensates. Phase separation is a mechanism where multivalent, low-affinity interactions lead to demixing of proteins or nucleic acids into dense, but still fluid, condensates [6,7]. RIM and RIM-BP were first identified to form condensates in vitro [8]. These condensates were competent to cluster voltage-gated calcium channels, a key function of RIM and RIM-BP [9], and possess plausible interactions with synaptic vesicles in vitro [10]. We recently showed *Caenorhabditis elegans* SYD-2/Liprin-α and ELKS also formed condensates, and that in vivo, SYD-2 and ELKS phase separation activity was critical for active zone assembly [11]. SYD-2 and ELKS condensates acted to robustly assemble active zone components during a transient liquid state during synaptogenesis. Correspondingly, homologous mammalian Liprin-αs have been shown to phase separate [12,13], with Liprin-α3 phase separation linked to active zone structure.

Beyond the presynaptic active zone, synapses contain additional condensate compartments. Synapsin and Synaptophysin form condensates on synaptic vesicles and contribute to synaptic vesicle clustering [14,15]. Condensates have also been observed at sites of ultrafast endocytosis adjacent to the active zone [16]. Further, in postsynapses, multiple components including PSD-95, SynGAP [17], and Rapsyn [18] have been identified to phase separate. These observations indicate phase separation is a common organizational mechanism in synapses and is critical for their formation [19] and function [20].

The mounting evidence for phase separation in synapses is contrasted with how little is known of the regulation of synaptic phase separation in vivo. It is presumably critical to form pre- and postsynaptic condensates at specific developing synaptic junction sites and regulate their properties to achieve a functional structure. Here, we identify phosphorylation as a mechanism controlling SYD-2/Liprin-α phase separation and presynaptic active zone formation. We identify the SAD-1 kinase to phosphorylate SYD-2 and determine phosphorylation controls an intramolecular autoinhibitory interaction. We find SAD-1 is localized to nascent synapses to activate presynaptic assembly during development through SYD-2 phase separation.

## Results

### Phosphorylation regulates SYD-2 phase separation and presynapse formation

Posttranslational modifications, including phosphorylation, regulate a variety of phase-separated condensates in cells [21]. To determine if phosphorylation could be regulating presynaptic active zone condensates formed by SYD-2/Liprin-α, we first immunoprecipitated endogenous GFP-SYD-2 from *C. elegans* and probed for serine and threonine phosphorylation. We find SYD-2 is indeed phosphorylated (Fig 1A), consistent with proteome-wide phosphorylation datasets [22]. We sought to test if SYD-2's phosphorylation regulates its phase separation and subsequent scaffolding functions in presynapse assembly. We endogenously fused a promiscuous lambda phosphatase domain (λpptase) to SYD-2's C-terminus to constitutively dephosphorylate the protein. This fusion tag effectively removed phosphorylation

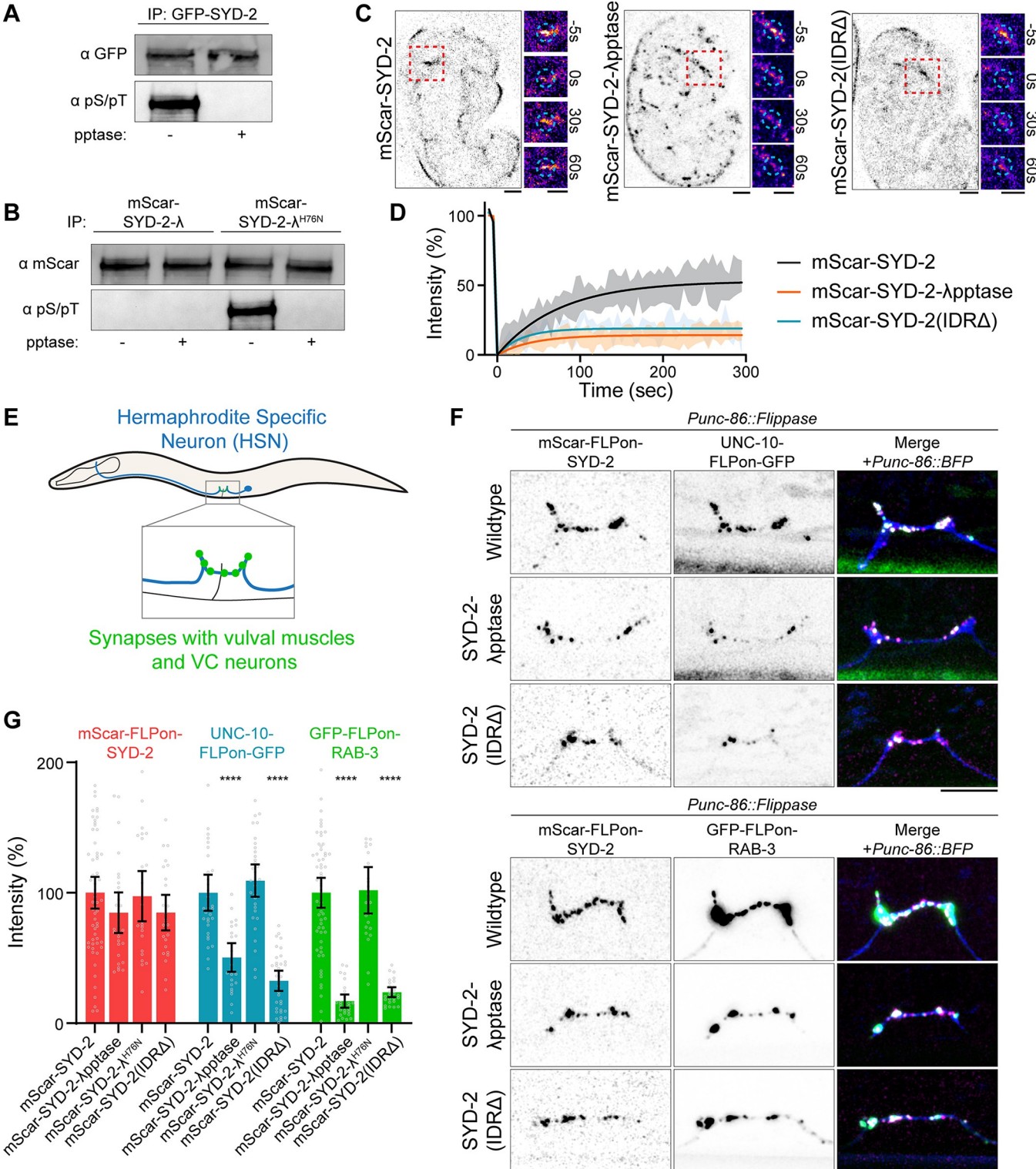

**Fig 1. Phosphorylation regulates SYD-2/Liprin-α's phase separation and active zone formation.** (A, B) Immunoprecipitation of endogenous GFP-SYD-2 (A) and endogenous mScarlet-SYD-2-λpptase (B). Western blot for GFP or mScarlet and phosphoserine/phosphothreonine. The indicated immunoprecipitation samples were treated with phosphatase before running on the gel (pptase +). (C) FRAP of endogenous mScarlet-SYD-2 at embryonic nerve ring synapses to measure dynamics. Wild-type SYD-2 is present in dynamic phase-separated condensates, while removal of phase separation in SYD-2 (IDRΔ) or removal of phosphorylation in SYD-2-λpptase inhibits condensate formation. Scale bars, 5 μm. (D) Quantification of FRAP in (C). (E) Schematic of

the *C. elegans* HSN that makes stereotyped synapses to vulval muscles to control egg laying. (F) HSN synapse formation phenotypes visualized with Airyscan superresolution imaging of endogenous fluorescent tags in the indicated mutants. Scale bar, 5 μm. (G) Quantification of HSN intensity in (F). ****, $p < 0.0001$. Underlying data is available in S1 Data. FRAP, fluorescence recovery after photobleaching; HSN, hermaphrodite-specific neuron.

from SYD-2 (Fig 1B), while fusion with a catalytically inactive λpptase$^{H76N}$ did not change SYD-2's native phosphorylation state. With this λpptase allele, we tested the impact of dephosphorylation on SYD-2 phase separation in vivo with fluorescence recovery after photobleaching (FRAP) assays at nascent embryonic synapses (Fig 1C). Because of the small size of presynaptic active zones, we are limited to photobleaching SYD-2 at an entire synapse. SYD-2 has been shown to be in a liquid condensate state at nascent synapses [11]; FRAP of an entire synapse therefore is a readout of the exchange between the SYD-2 condensate and cytoplasmic or other synaptic pools, measuring condensate liquidity. Wild-type SYD-2 at nascent synapses recovers quickly after photobleaching, consistent with liquid condensate formation [11]. These dynamics are lost when SYD-2's intrinsically disordered region (IDR) responsible for phase separation is removed [11]. The SYD-2-λpptase fusion showed a similar phenotype, with a decrease in FRAP dynamics reflecting inhibition of liquid condensate formation (Fig 1C and 1D).

To next determine the impact of phosphorylation on SYD-2's functions in presynapse formation that depend on phase separation, we imaged synapse formation in the *C. elegans* hermaphrodite-specific neuron (HSN). HSN accumulates synapses during late larval development that control egg laying (S1 Fig) [23,24]. We imaged cell-specific, endogenously tagged UNC-10/RIM, a downstream core active zone component [25], and RAB-3, a synaptic vesicle marker [26] (Fig 1E). The SYD-2-λpptase fusion caused a reduction in synaptic UNC-10 and failed to build large RAB-3 synaptic vesicle pools (Fig 1F and 1G). Intriguingly, SYD-2 levels remained normal at these synapses—a phenotype consistent with the complete loss of phase separation mutant, SYD-2(IDRΔ), which localizes normally but fails to build robust synapses (Fig 1F and 1G). The fact that SYD-2 accumulation does not change in these mutants also confirms that overall neuronal development has not been perturbed. A catalytically inactive λpptase$^{H76N}$ fusion had no impact on SYD-2 function or synapse formation, indicating that the fusion with λpptase does not itself perturb SYD-2 (Figs 1G and S2). These phenotypes therefore suggest phosphorylation of SYD-2 is critical to enable its phase separation and build a presynapse.

## The SAD-1 kinase phosphorylates SYD-2 to activate phase separation and presynapse assembly

A variety of kinases and signaling pathways have been implicated in synapse formation [27]. One candidate that stood out for possible regulation of SYD-2 is the SAD-1 kinase, which has been linked to neuronal development in vertebrates [28–30] and *C. elegans* [31–33]. When we imaged HSN synapse formation in a *sad-1Δ* mutant, we find a similar phenotype to SYD-2 (IDRΔ) or SYD-2-λpptase alleles (Fig 2A and 2B). SYD-2 localizes normally, but downstream UNC-10 and synaptic vesicle accumulation are reduced. In addition, we find that overexpression of constitutively active SAD-1(T202E) increases the accumulation of SYD-2, downstream active zone components, and synaptic vesicle marker RAB-3 at synapses (Figs 2A and 2B and S3). SAD-1(T202E) overexpression was performed with an *egl-6* promoter, chosen to achieve delayed expression in HSN to avoid major polarization and axon guidance defects associated with earlier overexpression [32]. These data argue that *sad-1* is a key regulator of presynapse accumulation during synaptogenesis. Further, these synapse formation phenotypes are consistent with a role in activating SYD-2 phase separation. Indeed, in a *sad-1Δ* mutant, SYD-2 condensate formation at nascent embryonic synapses is inhibited, as reflected by diminished

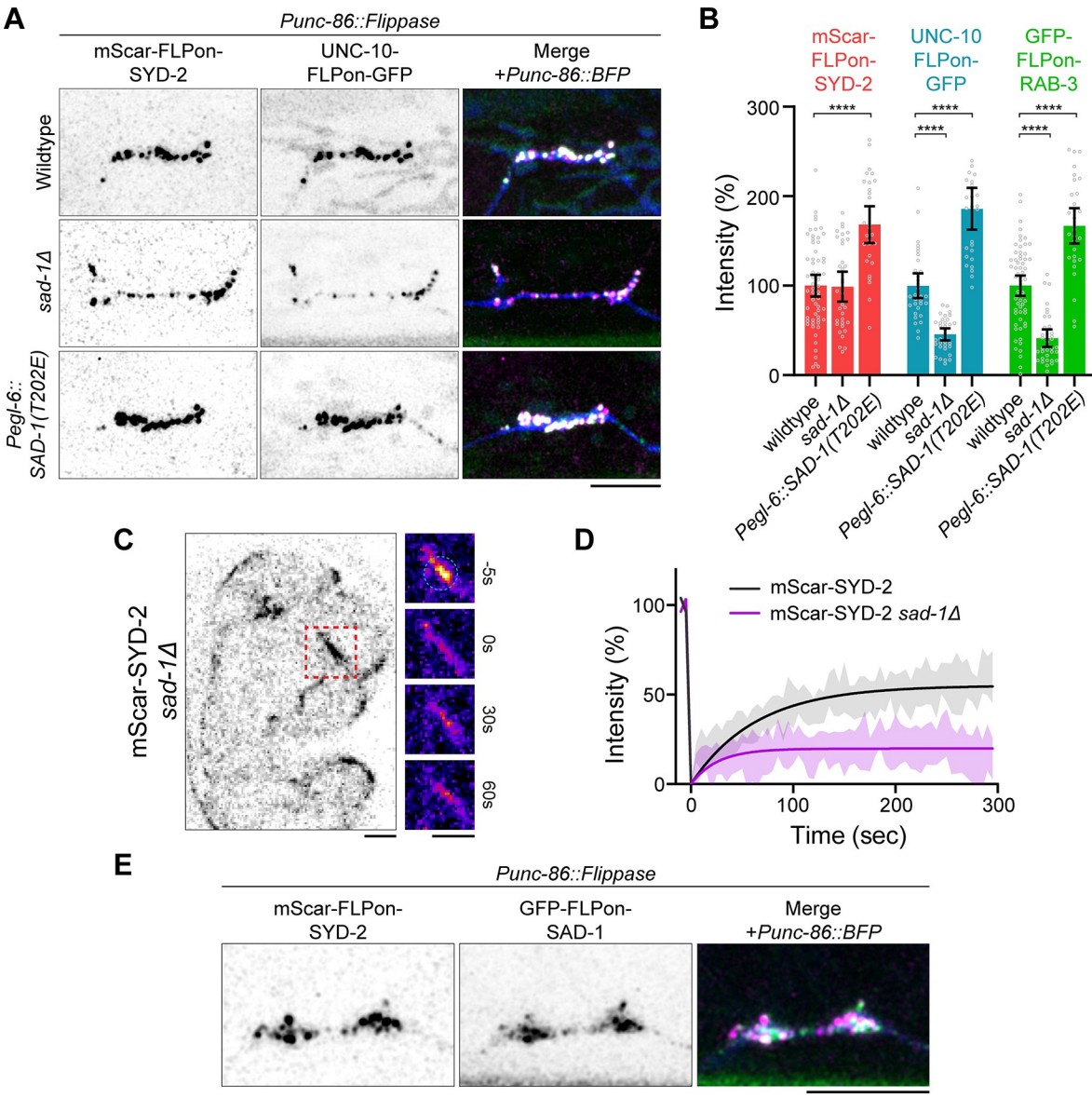

**Fig 2. The SAD-1 kinase regulates SYD-2/Liprin-α phase separation and active zone formation.** (A, B) HSN synapse formation phenotypes visualized with Airyscan superresolution imaging of endogenous fluorescent tags in the indicated mutants. *Pegl-6::SAD-1 (T202E)* is the expression of constitutively active SAD-1 kinase in HSN. Scale bar, 5 µm. (B) Quantification of HSN intensity in (A). ****, $p < 0.0001$. (C) FRAP of mScarlet-SYD-2 in a *sad-1Δ* background at embryonic nerve ring synapses to measure dynamics. Wild-type SYD-2 is present in dynamic phase-separated condensates and loss of *sad-1* inhibits condensate formation. Scale bars, 5 µm. (D) Quantification of FRAP in (C). (E) Airyscan superresolution image of endogenous GFP-SAD-1 at HSN presynapses. Scale bar, 5 µm. Underlying data is available in S1 Data. FRAP, fluorescence recovery after photobleaching; HSN, hermaphrodite-specific neuron.

FRAP dynamics (Fig 2C and 2D). We endogenously tagged SAD-1 with GFP and imaged its localization in HSN and find SAD-1 localizes distinctly to presynaptic sites (Fig 2E), positioning it appropriately to regulate presynapse formation and SYD-2. These results suggest SAD-1 signaling, either directly or indirectly, could be responsible for SYD-2 condensate phosphoregulation.

Despite SAD-1's implication in synapse formation and neuronal polarity, its substrates to accomplish these functions are not known. To identify possible substrates of SAD-1, we

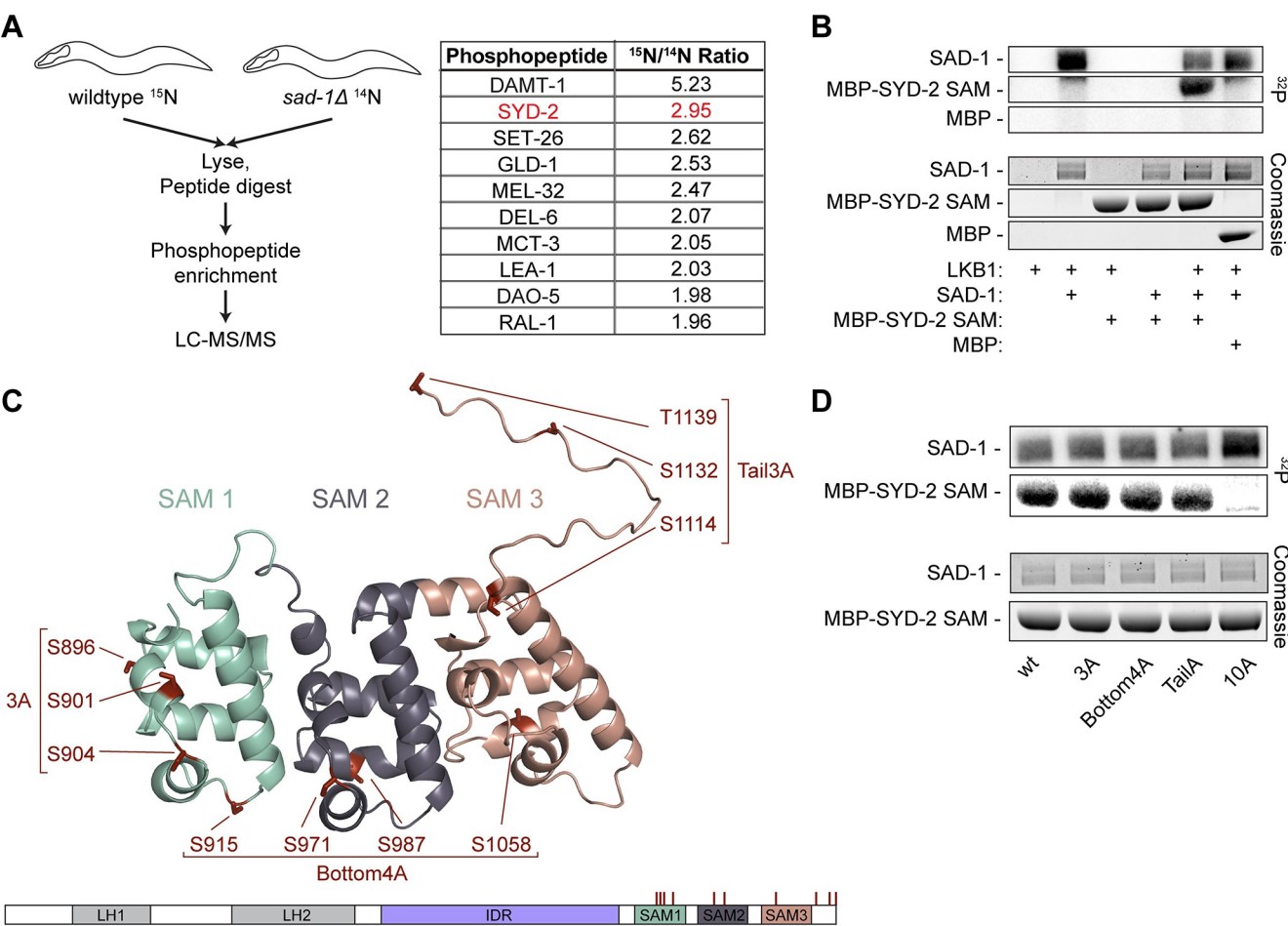

**Fig 3. SAD-1 kinase phosphoproteomics reveal direct phosphorylation of SYD-2.** (A) In vivo SAD-1 phosphoproteomics experiment to identify possible substrates. Protein names for the top 10 phosphopeptides identified are shown. See S1 Table for complete data. (B) In vitro kinase assay between SAD-1 and SYD-2's C-terminal SAM domains. SAD-1 is activated by the LKB-1 kinase complex. (C) Alphafold 2 model of SYD-2's SAM domains, residues 863–1139. Potential phosphosites identified from the phosphoproteomics in (A) and in vitro kinase assay in (B) are shown. Candidate phosphosites are organized into groups based on location for follow-up testing. See also S4A Fig and S1 and S2 Tables. (D) In vitro kinase assay between SAD-1 and SYD-2 phosphomutants. SAD-1 is activated by the LKB-1 kinase complex.

performed a phosphoproteomics screen (Fig 3A). Wild-type animals were labeled with heavy $^{15}$N and mixed with *sad-1Δ* $^{14}$N animals, before lysis, protein digestion, and phosphopeptide enrichment. Phosphopeptides were identified and quantified by LC-MS/MS and those with high $^{15}$N/$^{14}$N ratios represented potential SAD-1 substrates (Fig 3A). A variety of candidate substrates were discovered that may be phosphorylated by SAD-1 in vivo (Fig 3A and S1 Table). Fortuitously, a SYD-2 phosphopeptide was a top hit present in all 3 biological replicates (Fig 3A), indicating SAD-1 may be responsible for SYD-2 phosphoregulation.

To test if SAD-1 was directly responsible for phosphorylating SYD-2 and determine if additional sites may be present that were not found in the phosphoproteomics screen, we performed in vitro kinase assays between SAD-1 and SYD-2 (Figs 3B, S4A, and S4B and S2 Table). Purified recombinant SAD-1 kinase was pre-activated through phosphorylation by the LKB-1 complex [31] and mixed with purified SYD-2 fragments. Activated SAD-1 was effective at phosphorylating SYD-2's SAM and IDR domains in vitro (Figs 3B, S4A, and S4B and S2 Table). We mapped sites from these assays with mass spectrometry to identify 10 sites on SYD-2's SAM domains and 26 sites in SYD-2's IDR (Figs 3C and S4A).

We anticipated that not all of the sites identified from in vitro phosphorylation assays would represent true in vivo phosphosites, as in vitro kinase conditions are generally promiscuous. Only a few phosphosites were seen in our in vivo phosphoproteomics or in previous whole-proteome datasets (S4A Fig) [22]. In vitro, however, these sites were robust and only mutation of all 10 sites in the SAM domains (Fig 3D) or 26 sites in the IDR (S4B Fig) was capable of abolishing phosphorylation. We therefore separated sites into groups based on structural location to evaluate in vivo function (Figs 3C and S4A). We introduced alanine mutations at each group of sites into the endogenous SYD-2 gene and imaged HSN synapse formation to assay function. The 26 sites in SYD-2's IDR, including 12 sites clustered in a key phase separation motif [11], showed no evidence of presynaptic function, with normal synapses formed in HSN (S4C and S4D Fig). IDR phosphomutants also showed no synaptic transmission defects in an assay for cholinergic synaptic transmission (S4E Fig). We conclude IDR phosphorylation by SAD-1 is not required for SYD-2's function and may be an artifact of in vitro phosphorylation conditions.

A 3A (S896A, S901A, S904A) mutation of sites clustered in the most N-terminal SAM domain, however, showed a significant synapse formation phenotype (Figs 4A, 4B, and S5A), similar to SYD-2-λpptase or *sad-1Δ*, with diminished UNC-10 and RAB-3 assembly and normal SYD-2 levels. Other clusters of sites on the bottom face or tail of SYD-2's SAM domains did not impact synapse formation. A 10A mutant, encompassing all SAM domain sites identified in vitro, reproduced the synapse formation defects of the 3A mutant, but also impacted SYD-2 overall levels at synapses, perhaps due to impacts on protein stability. In sum, 3 sites (S896, S901, and S904) show functional impacts on synapse formation. Consistently, the 3A phosphomutant of these sites also showed synaptic transmission defects, similar to the complete loss of phase separation IDRΔ mutant (Fig 4C), indicating synapses body-wide are impaired in the absence of this phosphorylation. Intriguingly, these 3 sites are well conserved in human and *Drosophila* Liprin-α homologs (S5B Fig).

To test whether these phenotypes result from the regulation of SYD-2's phase separation, we assayed SYD-2 phosphomutant dynamics at developing synapses with FRAP. The 3A mutant showed a significant loss of dynamics (Fig 4D and 4E), indicating inhibition of phase separation, while other SAM phosphomutants had no effect.

These phenotypes suggest 3 sites in SYD-2's SAM domains are critical for regulating its phase separation and presynaptic function. To determine if these SYD-2 phosphosites account for SAD-1's function in promoting synapse formation, we extended the overexpression of constitutively active SAD-1(T202E) into an SYD-2(3A) background. In the 3A phosphomutant, SAD-1 was not able to increase levels of SYD-2, UNC-10, and synaptic vesicles at HSN presynapses (Fig 5A and 5B), as seen in a wild-type background. These results indicate SAD-1 activates presynaptic active zone formation through the phosphorylation of 3 sites within SYD-2's SAM domain.

Since blocking phosphorylation inhibited SYD-2 phase separation and synapse formation, we next sought to test if phosphomimetic mutations or the removal of SAM domains containing these regulatory elements could constitutively activate SYD-2 phase separation and synapse formation. However, we find SYD-2(3E) and SYD-2(SAM1-3Δ) mutations result in a strong depletion of SYD-2 protein at HSN synapses (Fig 6A and 6B) and a loss-of-function phenotype (S6A and S6B Fig). We considered that constitutive phase separation could be occurring instead in the neuron's cell body, preventing synaptic localization. We imaged SYD-2 in the HSN cell body and found a generally diffuse cytoplasmic localization with occasional puncta; however, we detected no differences between wild-type, 3E, and SAM1-3Δ mutants. Finally, we reasoned that precocious phase separation may be causing degradation in the cell body, preventing accumulation or localization to synapses. To test this, we treated animals with bortezomib, a proteasome inhibitor, and imaged SYD-2 in the cell body (Fig 6C). Wild-type SYD-2 levels did not change upon this treatment, but SYD-2(3E) and SYD-2(SAM1-3Δ) mutants increased in intensity and localized

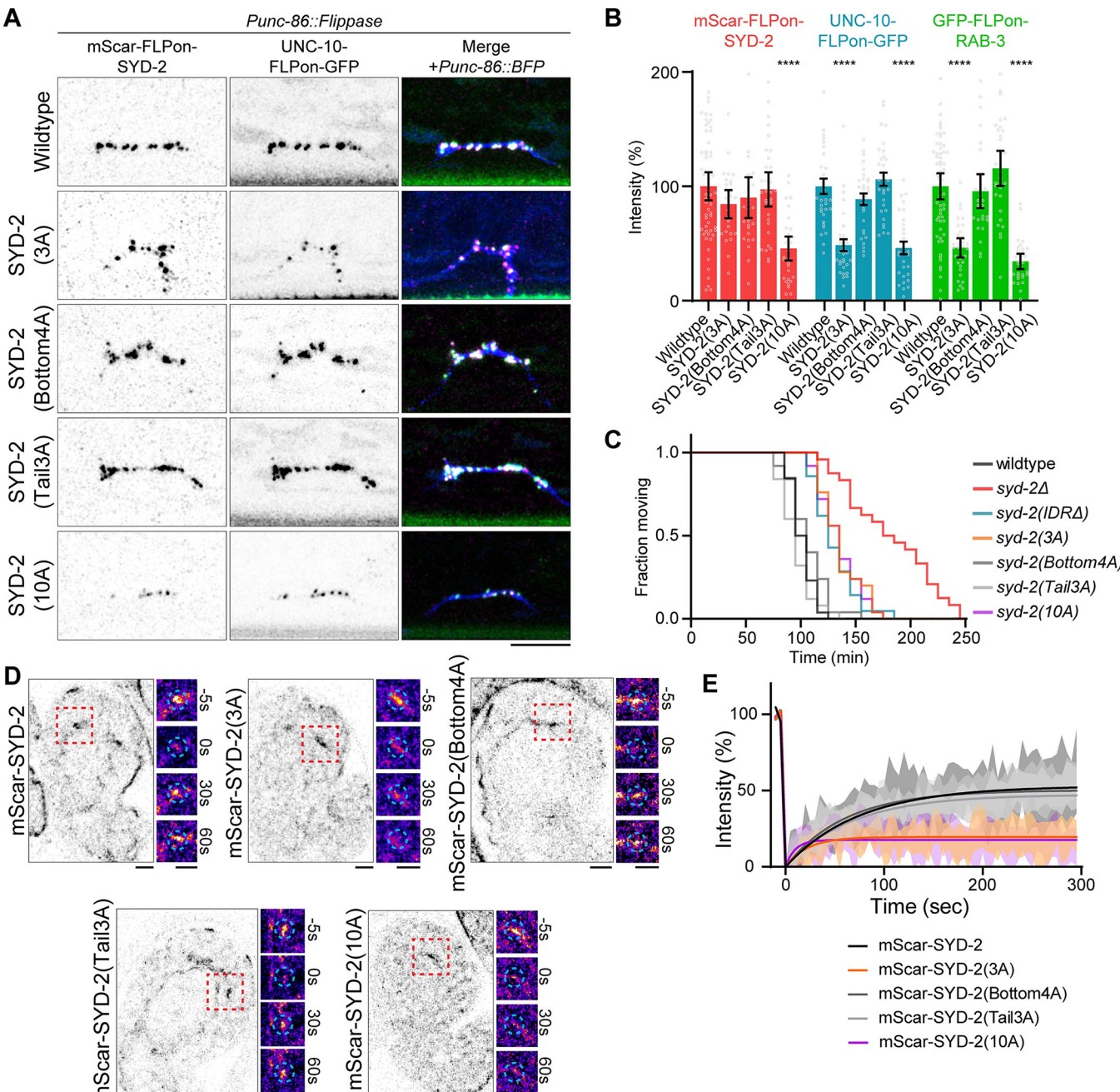

**Fig 4. SAD-1 phosphorylation of SYD-2 at 3 sites controls phase separation and active zone assembly.** (A) HSN synapse formation phenotypes visualized with Airyscan superresolution imaging of endogenous fluorescent tags in the indicated mutants. Scale bars, 5 μm. (B) Quantification of HSN intensities in (A). ****, $p < 0.0001$. (C) Aldicarb synaptic transmission assay. Extended time to paralysis on 1 mM Aldicarb indicates defective synaptic transmission. $n > 20$ for each genotype. (D) FRAP of mScarlet-SYD-2 phosphomutants at embryonic nerve ring synapses to measure dynamics. Wild-type SYD-2 is present in dynamic phase-separated condensates, while certain phosphomutants inhibit condensate formation. Scale bars, 5 μm. (E) Quantification of FRAP in (D). Underlying data is available in S1 Data. FRAP, fluorescence recovery after photobleaching; HSN, hermaphrodite-specific neuron.

increasingly to puncta (Fig 6D and 6E). Bortezomib treatment had no effect on synapse-localized SYD-2 (S6C and S6D Fig). This result indicates that SYD-2(3E) and SYD-2(SAM1-3Δ) are being degraded in the cell body. When degradation is blocked, the mutants accumulate in puncta, consistent with precocious phase separation due to loss of regulation.

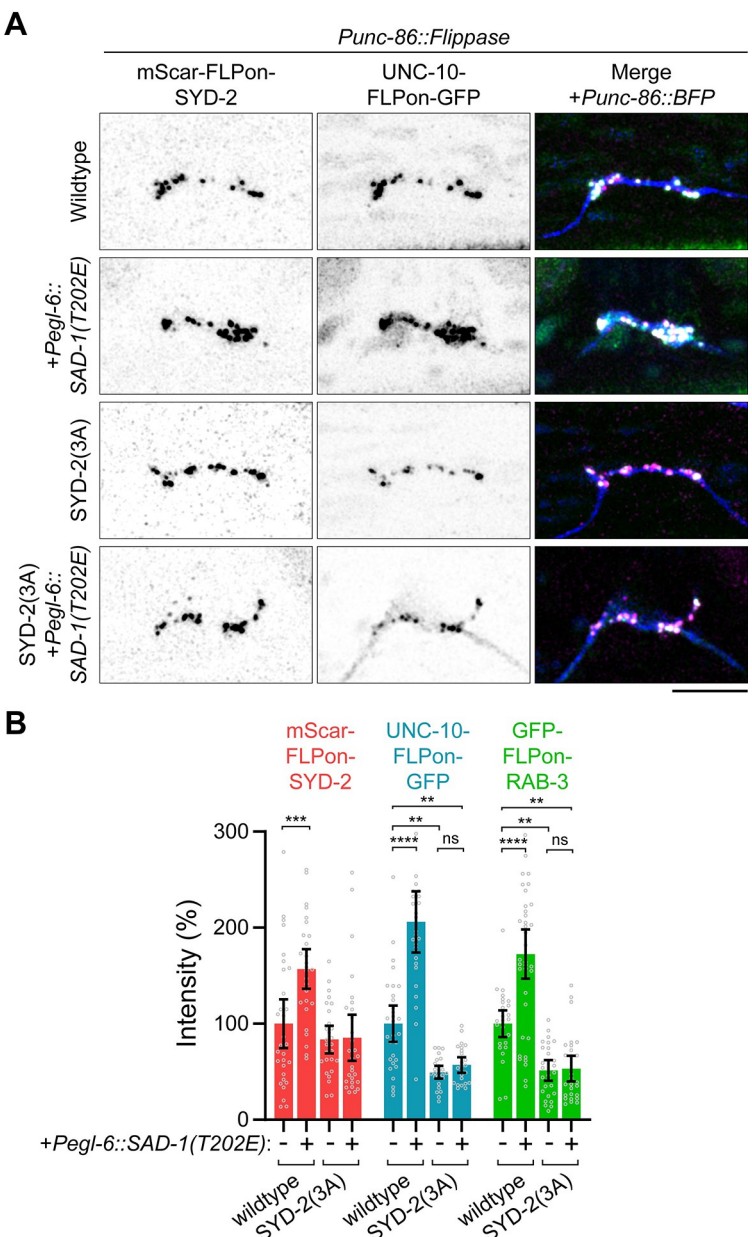

**Fig 5. SAD-1 control of active zone assembly requires SYD-2 phosphorylation.** (A) HSN synapse formation phenotypes visualized with Airyscan superresolution imaging of endogenous fluorescent tags in the indicated mutants. *Pegl-6::SAD-1(T202E)* is an overexpression of constitutively active SAD-1 kinase in HSN. Scale bar, 5 μm. (B) Quantification of HSN intensities in (A). **, $p < 0.01$; ***, $p < 0.001$; ****, $p < 0.0001$; ns, not significant. Underlying data is available in S1 Data. HSN, hermaphrodite-specific neuron.

## Phosphorylation of SYD-2 activates phase separation by relieving autoinhibition

In previous examples of phase separation regulation by phosphorylation, direct phosphorylation of IDRs is thought to alter phase separation properties [21]. Here, we find phosphorylation in a folded domain adjacent to an IDR to be responsible for modulating phase separation. We hypothesized that these phosphosites may instead regulate phase separation through

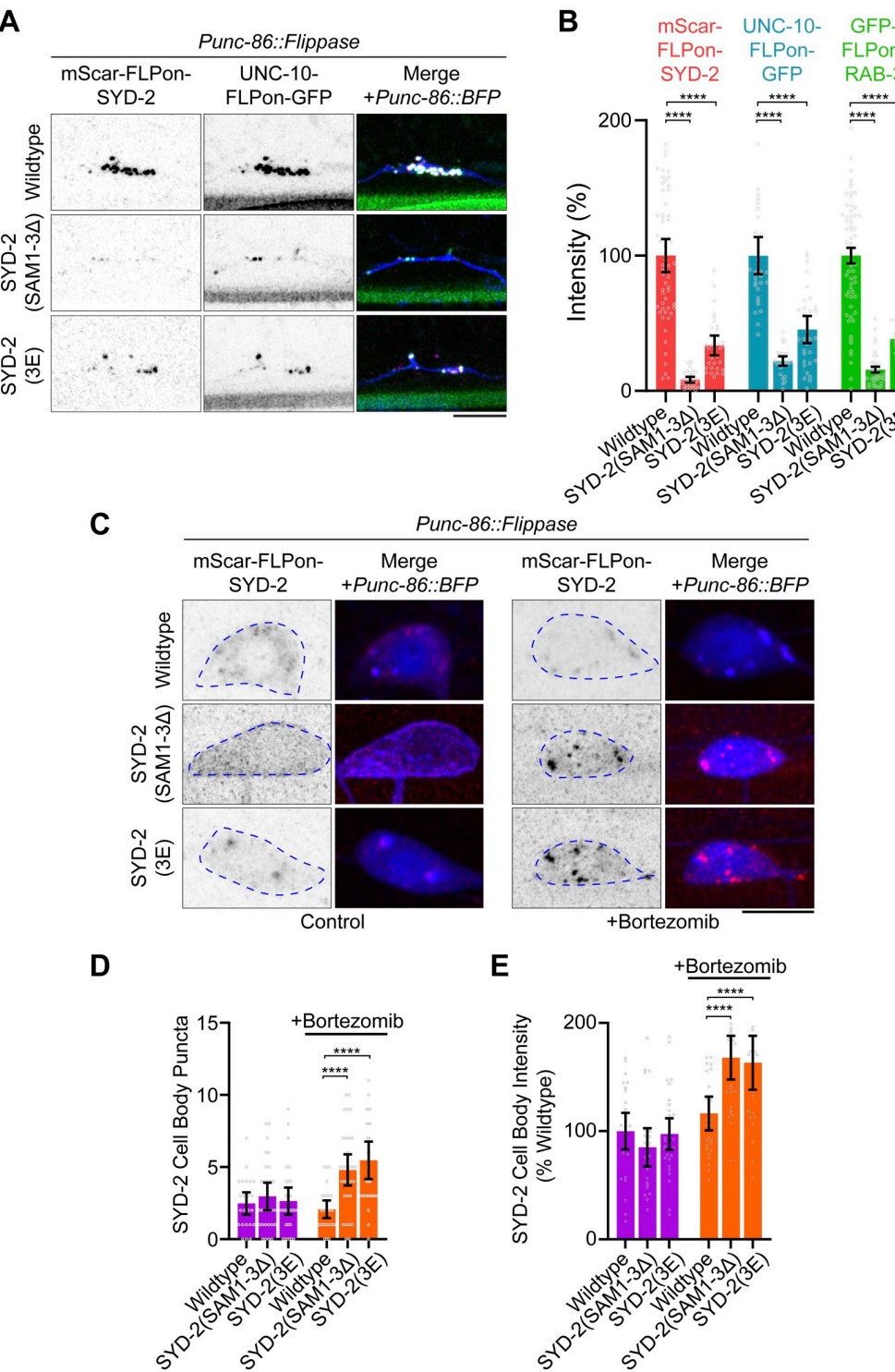

**Fig 6. Constitutive activation of SYD-2 leads to precocious aggregation and degradation.** (A) HSN synapse formation phenotypes visualized with Airyscan superresolution imaging of endogenous mScarlet-SYD-2 and indicated mutants. SAM1-3Δ and 3E mutants are depleted at HSN synapses. Scale bar, 5 μm. (B) Quantification of HSN intensities in (A). (C) HSN cell body imaging of endogenous mScarlet-SYD-2 and indicated mutants; 10 μm of the proteasome inhibitor bortezomib or DMSO as a control was added to the indicated animals for 4 h prior to imaging. Scale bar, 5 μm. (D) Quantification of SYD-2 puncta in (C). ****, $p < 0.001$. (D) Quantification of total SYD-2 cell body intensity in (C). ****, $p < 0.001$. Underlying data is available in S1 Data. HSN, hermaphrodite-specific neuron.

conformational changes or intramolecular interactions. To test this in vitro, we split the SYD-2 protein in two (Fig 7A), into an N-terminal fragment containing coiled-coils and the IDR (Nter), and a C-terminal fragment containing the SAM domains (SAM). We confirmed the Nter fragment was capable of phase separation, as shown previously [11], and found the SAM fragment alone was not (Fig 7B). At high concentrations, the addition of the SAM domains to the Nter condensates results in their incorporation, supporting a possible interaction between the two (Fig 7B). However, we noted at lower concentrations, addition of SAM domains prevented Nter phase separation. To quantify the impact of SAM domains on Nter phase separation, we characterized phase diagrams for SYD-2's Nter alone versus Nter + SAM condensates. We find the addition of SAM domains robustly increased the critical concentration of Nter droplet formation (Fig 7C). Interestingly, the Nter + SAM droplets formed at high concentration were not dynamic (Fig 7D), and a SAM(3E) phosphomimetic construct was able to partially rescue these dynamics. Together, these data indicate SYD-2's SAM domains inhibit the protein's ability to phase separate.

These results led us to consider that SYD-2's SAM domains might interact with its Nter in order to inhibit phase separation. We hypothesized that phosphorylation may release this inhibitory binding to enable phase separation. To test this model, we performed in vitro binding assays between SYD-2 N-terminal fragments and SAM domains (Fig 7E). We find that SYD-2's SAM domains directly interacted with an Nter fragment containing the IDR, but an IDR fragment alone was not sufficient for the interaction. We therefore conclude that the SAM domains interact with the coiled-coil regions in SYD-2's N-terminus. We tested if phosphorylation modulates this interaction by pre-phosphorylating SYD-2's SAM domains with the SAD-1 kinase. SAD-1 phosphorylation inhibited SAM domain interaction with the N-terminal fragment. This inhibition specifically required the 3 phosphosites we determined functioned in synapse formation (Fig 4), as an SAD-1-phosphorylated SAM(3A) construct restored binding to the N-terminus. We conclude SYD-2's SAM domains directly interact with its N-terminus to inhibit phase separation, which is released by specific SAD-1 phosphorylation.

Together, our results support a model for intramolecular regulation of IDR-mediated phase separation. SYD-2's unphosphorylated C-terminal SAM domains bind in the N-terminus, inhibiting the intervening IDR's phase separation. Intriguingly, this mechanism may be agnostic to the intermediate IDR, as inhibitory binding occurs between 2 IDR-flanking domains. To test the generalizability of the intramolecular autoinhibition, we introduced the 3A phosphosite mutation into a previously characterized allele that replaced SYD-2's IDR with the IDR of FUS (Fig 7F). Swapping SYD-2's IDR for FUS's rescues its phase separation and synapse formation functions [11]. The addition of the 3A mutation inhibited synapse formation in SYD-2 (IDRΔ+FUS), indicating these sites and the intramolecular interactions they control are capable of regulating the phase separation of generic IDR motifs in the protein (Fig 7F and 7G). Thus, SAD-1 phosphorylation controls an autoinhibitory interaction between N- and C-terminal domains in SYD-2 that inhibit phase separation of an intermediate IDR.

## SAD-1 is poised at nascent synapses to activate active zone phase separation

Our data suggest SYD-2 phosphorylation by SAD-1 activates the protein to phase separate and assemble the presynaptic active zone. Indeed, SAD-1 localizes strongly to presynaptic sites, as determined by an endogenous GFP-SAD-1 allele (Fig 2E). To determine if this localization is dependent on SYD-2 and the active zone or an upstream synaptic adhesion molecule, we imaged SAD-1 localization in *syd-2Δ* and *syd-1Δ* [25,34] mutants. We find that SAD-1

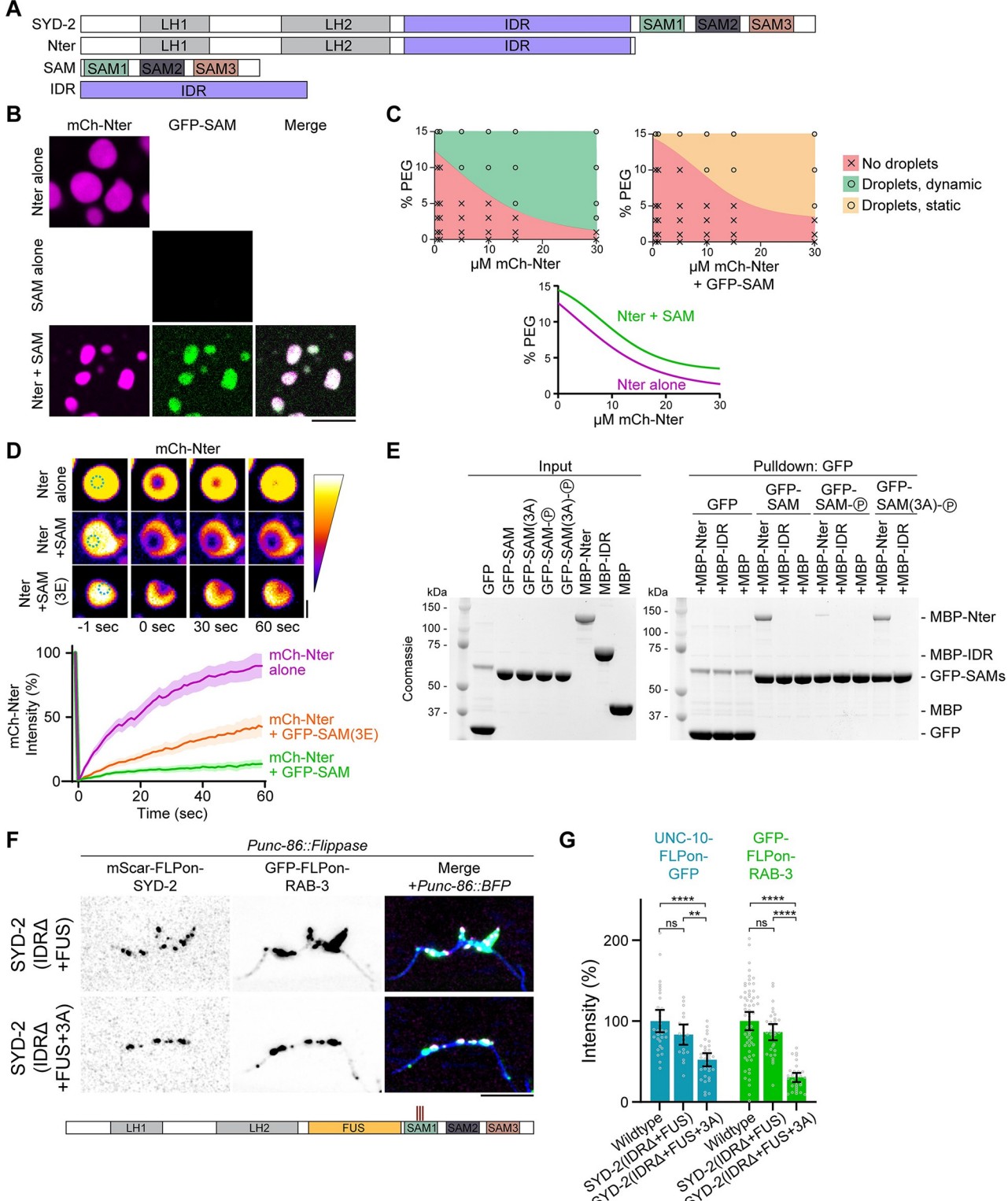

**Fig 7. SAD-1 phosphorylation relieves SYD-2 autoinhibition to activate phase separation.** (A) Domain schematic of SYD-2 and constructs used in this figure. (B) In vitro phase separation of SYD-2 Nter and SAM domains alone and together at 10 μm. Dilution to 150 mM NaCl and addition of 10% PEG 3000 were used to induce phase separation. Scale bar, 5 μm. (C) Phase diagrams of SYD-2 Nter and Nter+SAM, constructed from phase separation assays at each protein and PEG 3000 concentration. Bottom: Phase boundaries extracted from the phase diagrams above. Addition of SAM domains increased the concentration and crowding required for phase separation, indicating inhibition of Nter phase separation. (D) Fluorescent

recovery after photobleaching measuring internal dynamics of mCh-SYD-2(Nter) condensates with and without the addition of SAM domains. Performed at 10 μm Nter and SAM and 10% PEG 3000. Quantification below. Addition of SAM domains to SYD-2 Nter condensates inhibits dynamics of in vitro condensates. A SAM(3E) phosphomimetic partially rescues this inhibition. Scale bar, 1 μm. (E) In vitro binding assay between SYD-2 domains. Constructs labeled "(P)" were pre-phosphorylated by SAD-1 prior to the binding assay. Binding is specific between Nter and SAM domains and is inhibited by phosphorylation of 3 sites (S896, S901, and S904). (F) HSN synapse formation phenotypes visualized with Airyscan superresolution imaging of the indicated mutants. SAD-1 phosphosites are capable of regulating phase separation of an FUS replacement construct, where SYD-2's IDR is replaced by an FUS phase separation motif. Scale bar, 5 μm. (G) Quantification of HSN intensity from (F). ****, $p < 0.0001$; ns, not significant. Underlying data is available in S1 Data. HSN, hermaphrodite-specific neuron; IDR, intrinsically disordered region.

localization does not strictly require either *syd-2* or *syd-1* and active zone formation: SAD-1 is decreased, but generally remains localized to presynaptic sites when these genes are absent (Fig 8A and 8B). However, in a *syg-1Δ* mutant, a synaptic cell adhesion molecule responsible for establishing the HSN synaptic region and initiating synapse formation [35,36], SAD-1 localization is lost (Fig 8A and 8B). Therefore, SAD-1 is positioned at presynaptic sites through upstream synaptic cell adhesion cues where it can phosphorylate and activate SYD-2's phase separation to build the presynaptic active zone (Fig 8C).

## Discussion

In this study, we have identified the SAD-1 kinase to activate presynaptic SYD-2/Liprin-α phase separation and active zone assembly. SAD-1 has been implicated in various neuronal development processes, and here we find specific loss-of-function and overactivation phenotypes for SAD-1 in presynaptic active zone assembly. We identify a key SAD-1 substrate in SYD-2/Liprin-α that accounts for SAD-1's function in presynaptic assembly. SYD-2 was previously shown to phase separate during presynaptic active zone assembly; we find SAD-1 phosphorylation is a key regulatory mechanism that activates SYD-2's phase separation function at nascent synapses.

The group of phosphorylation sites we identify as responsible for SYD-2 regulation is well conserved with mammalian and *Drosophila* Liprin-α homologs of SYD-2. Serine 896, 901, and 904 residues are conserved in *Drosophila* Liprin-α and all 4 human Liprin-αs, with the exception of an S901A change in Liprin-α3. In addition, phosphorylation has been observed at multiple of these sites in mammalian genome-wide phosphoproteomics datasets [37]. Considering this conservation, the functional conservation of phase separation [12], and the functional conservation of SYD-2/Liprin-α in active zone assembly [3,38], we consider it likely the regulation discovered here is conserved. These phosphorylation sites in SYD-2's SAM domains are nearby, but not within, known binding interfaces of CASK and Liprin-β [39]. The phenotypes seen here cannot be explained by alterations in these binding partners as *C. elegans* CASK and Liprin-β homologs are not critical for presynaptic development [40,41].

Previously, phosphorylation by PKC was also observed at S760 in Liprin-α3, which was seen to regulate its ability to phase separate [12]. This site was not, however, necessary for Liprin-α2 to phase separate. The S760 site is not explicitly conserved in SYD-2, though it is very near a number of IDR phosphosites we tested and found had no clear function (12A, S4 Fig). IDR sequences have been observed to diverge quickly at the sequence level, while presumably preserving function [42]. It will be necessary to test if other sites in SYD-2 are phosphorylated by PKC to substitute a similar regulation.

A key question in neuronal development is how intracellular assembly of synaptic structures is connected to adhesion molecule-based specificity mechanisms [5]. The SAD-1 activation signal we have identified likely occurs at nascent presynaptic sites where SAD-1 is strongly localized. Previously, SAD-1 has been seen to moderately depend on SYD-2 and SYD-1 for localization [25], placing it "downstream" in an assembly hierarchy. However, these

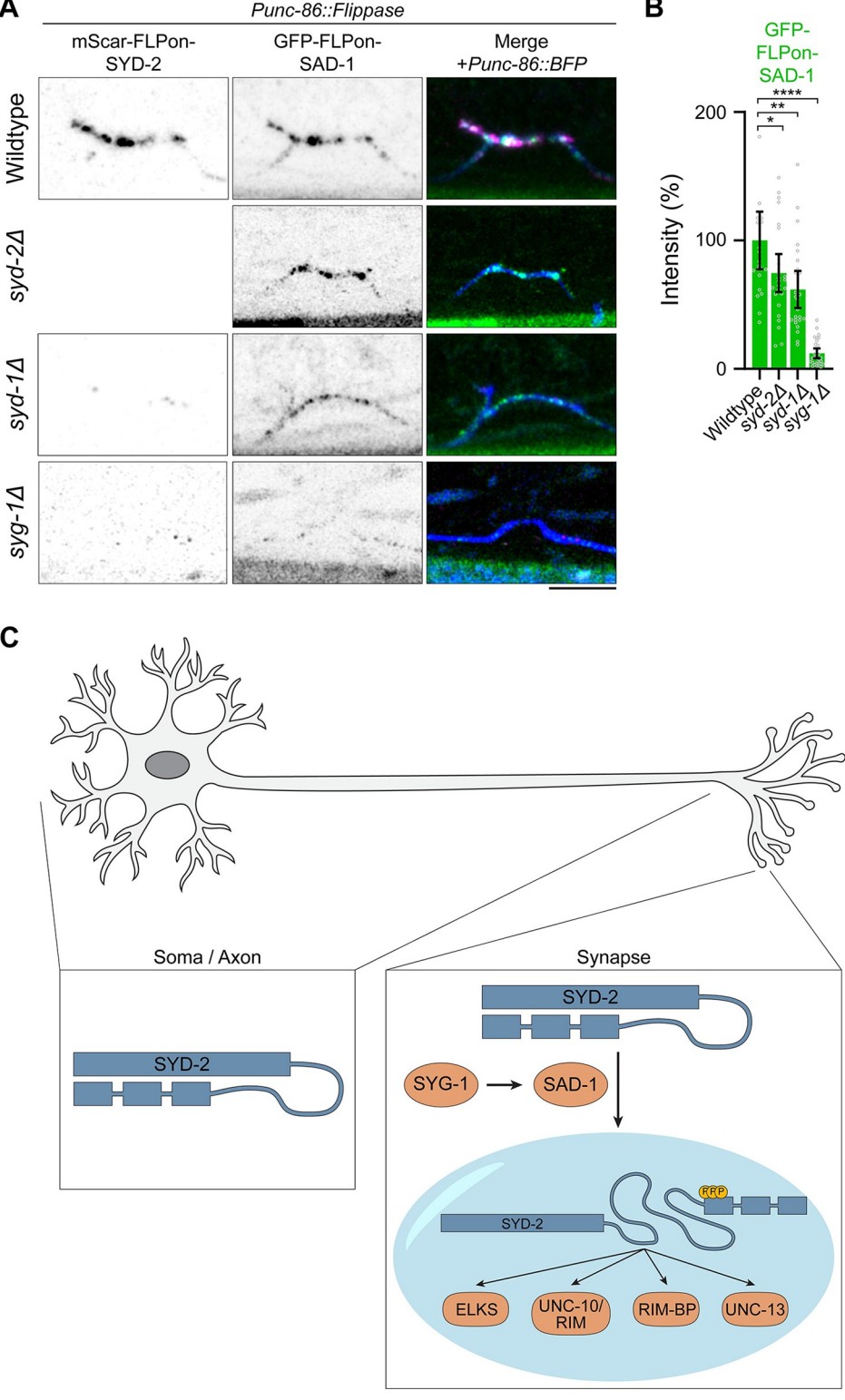

**Fig 8. SAD-1 localizes to presynapses upstream of the active zone in order to activate phase separation and active zone formation.** (A) Airyscan superresolution images of endogenous GFP-SAD-1 at HSN synapses with or without *syd-2*, *syd-1*, or *syg-1*. Scale bar, 5 μm. (B) Quantification of GFP-SAD-1 HSN intensity in (A). ****, $p < 0.0001$. (C) Model of SAD-1 activation of SYD-2 phase separation and active zone formation. Underlying data is available in S1 Data. HSN, hermaphrodite-specific neuron.

experiments were performed with overexpressed proteins and were not quantitative. With knock-in fluorescent tags, representing endogenous protein levels and localization, we see only a minor dependence of SAD-1 localization on SYD-2 and SYD-1; approximately 70% of SAD-1 protein remains localized to presynaptic sites in the absence of these proteins. We find that SAD-1 localization to presynapses is instead entirely dependent on SYG-1, an upstream synaptic cell adhesion molecule that instructs the location of synapses in HSN and select other neurons [35,43], likely through the formation of an actin network [44]. SAD-1 has been seen to bind NAB-1 [45], an actin-binding protein which also depends on SYG-1 for localized to presynaptic sites. Thus, the data suggests upstream synaptic cell adhesion molecule SYG-1 localizes SAD-1 to nascent synapses where it activates SYD-2 by phosphorylation of SAM domains.

In addition to SYD-2, we find a variety of possible SAD-1 substrates in our phosphoproteomics screen (S1 Table). Intriguingly, the substrates identified are enriched in proteins involved in signaling ($p = 1.5*10^{-8}$), cytoskeleton ($p = 1.8*10^{-7}$), and neuronal function ($p = 1.1*10^{-4}$). Additional study on these substrates may reveal links to SAD-1's described functions in neuronal polarity and neurite outgrowth. SYD-2 phosphorylation appears to account for SAD-1's specific function in presynapse assembly, as an SYD-2(3A) mutant phenocopies a *sad-1Δ* and SAD-1 overexpression is unable to impact synaptic accumulation in SYD-2(3A).

Beyond the neuronal development signaling revealed here, we have identified a novel mechanism that controls phase separation by an internal IDR. In most examples of phase separation regulation by posttranslational modifications, direct modification of IDRs alters biochemical properties that impact the ability to phase separate [21]. Here, instead, we find phosphorylation controls an intramolecular binding interaction. When dephosphorylated, binding between domains surrounding the IDR occurs. Phosphorylation releases this binding, while enabling IDR phase separation. Loss of this regulation in 3E or SAM1-3Δ mutants lead to aggregation and degradation in the cell body. Curiously, overexpression of SYD-2 constructs lacking SAM domains were seen to rescue *syd-2Δ* [46], indicating overexpression can overcome this dysregulation.

This mechanism is particularly intriguing for spatial regulation and activation of phase separation, as the newly translated, unphosphorylated protein will adopt an autoinhibited form. Only upon phosphorylation at the proper place, in this case nascent synaptic sites, is autoinhibition relieved to enable IDR phase separation. As SYD-2 condensates scaffold a variety of additional active zone components [3,11,25], autoinhibition is an attractive mechanism to prevent improper assembly at the wrong place and time in the neuron.

Curiously, this autoinhibition mechanism was agnostic to the IDR within SYD-2 performing the phase separation. How binding between folded domains flanking the IDR actually inhibits its phase separation is not yet clear. It is possible that the bound state "stretches" the IDR into an extended conformation that prevents the multivalent interactions underlying phase separation [6,7]. As more IDRs are identified to function in condensate formation, similar autoinhibitory and extra-IDR regulation may be at play due to the advantageous autoinhibition, allowing the cell to prevent condensate formation until needed.

## Methods

### *C. elegans* methods

*C. elegans* strains (S3 Table) were grown on OP50 *E. coli*-seeded nematode growth media (NGM) plates at 20°C, following standard protocols [47]. N2 Bristol is the wild-type reference strain. The following loss-of-function alleles were used in this study: *sad-1Δ*: *sad-1(ky289)*; *syd-2Δ*: *syd-2(wy5)*; *syd-1Δ*: *syd-1(ju82)*; *syg-1Δ*: *syg-1(wy652)*. [15]N wild-type worms for

phosphoproteomics were grown for at least 10 generations on $^{15}$N-labeled, MG1655 *E. coli*-seeded, nitrogen-free NGM plates at 20°C, as described previously [48]. $^{15}$N MG1655 was prepared in M9 minimal media containing $^{15}$NH$_4$Cl as the sole nitrogen source.

Nerve ring imaging was performed on comma-stage embryos. HSN imaging was performed on synchronized early L4 hermaphrodites, as determined by vulval morphology (S1 Fig). HSNL (left) was exclusively imaged due to its advantageous separation from the nerve cord. Aldicarb assays were performed on day 1 adult hermaphrodites on NGM plates containing 1 mM Aldicarb (Millipore-Sigma). Bortezomib was added to plates where indicated at 10 μm final concentration for 4 h before imaging. Arrays were created by gonadal microinjection. *Pegl-6*::*SAD-1(T202E)* arrays were injected at 30 ng/μl with a 50 ng/μl *Podr-1*::*GFP* coinjection marker. FLPon tags were constitutively flipped out by temporary expression of a germline *Peft-1*::*Flippase*.

## Constructs and CRISPR-Cas9 genome editing

Constructs (S4 Table) were created with an isothermal assembly (Gibson) method [49]. pNM171 *Punc-86*::*tagBFP2-SL2-FLP* was assembled into an empty pSK vector from the 5,102-nucleotide promoter of *unc-86*, a *C. elegans* codon-optimized tagBFP2 (pJJR81, Addgene #75029), an SL2 site, Flippase (pDML63 [50]), a self-excising Hygromycin resistance cassette [51], and flanking 500-bp homology arms to the MosSCI site ttTi5605 [52]. pNM172 *Pegl-6*::*SAD-1(T202E)* was assembled into a pSM delta vector from the 3,527-nucleotide promoter of *egl-6* and SAD-1 cDNA. A T202E activating mutation was introduced with site-directed mutagenesis, designed based on alignment to orthologous MAP kinase activation loops [53,54]. Lambda phosphatase from *Escherichia* phage lambda was synthesized as a *C. elegans* codon-optimized gBlock (IDT) and assembled into a pSK vector. A H76N inactivating mutation [55] was introduced with site-directed mutagenesis. SYD-2 SAM(844–1139), SYD-2 IDR(517–843), and SAD-1 constructs were assembled into pHis6-GFP, pHis6-mCherry, and pMBP-his empty vectors. 3A, 3E, Bottom4A, and Tail3A phosphomutants were introduced into SYD-2 constructs using site-directed mutagenesis. 10A, 12A, and 26A phosphomutants were synthesized as gBlocks (IDT) and assembled into pMBP-his vectors.

Endogenous genome modifications were created as previously described [11,56] through gonadal microinjection of Cas9 (IDT), tracrRNA (IDT), gRNA (IDT), and PCR repair template mixtures. mScarlet-I-FLPon cassettes used FRT(F3) sites (GAAGTTCCTATTCTTCA AATAGTATAGGAACTTC) and GFP-FLPon cassettes used FRT (GAAGTTCCTATTCTC TAGAAAGTATAGGAACTTC) sites for simultaneous compatibility. WySi974, a single-copy *Punc-86*::*tagBFP2-SL2-FLP* that drives Flippase and a tagBFP2 morphology marker in HSN, was created by insertion into the MosSCI ttTi5605 site on chromosome V as a convenient and well-established expression locus [52] using a hygromycin resistance strategy [51]. SYD-2 phosphomutants were introduced into SYD-2(IDRΔ) or SYD-2(SAM1-3Δ) strains with repair templates generated by PCR from pSK vectors or gBlocks, described above. The IDRΔ+FUS replacement was recreated in an mScarlet-I-FLPon-SYD-2 strain as previously described [11]. The following gRNAs were used for genome edits: mScar-FLPon-SYD-2: AGAAATATGAGCTACAGCAA; UNC-10-FLPon-GFP: GATTCCGATGTATCAGTTGG; SYG-1-FLPon-GFP: TGAGTTGATGT TCGACTAAT; Lambda phosphatase (wild type and H76N): TTTAATTTAACTAACTAACT; IDRΔ: GAACTGCGCAATTCCAGTCA and GGCGAGCAGTCGGGCACAGA; GFP-FLPon-SAD-1: CATGACTGCGCTCGTCAATC; SAMΔ: TCCAACTGTTGTTGCCTGGC and TTTAATTTAACTAACTAACT; SAM phosphomutants (3A, Bottom4A, Tail3A, 10A, 3E): AAT TACCAAGCAACAACAGT; IDR phosphomutants (12A, 26A): AATGCAAGAACTGCGCA ATT. All genome-edited strains were outcrossed and verified by sequencing.

## Protein methods

*C. elegans* were prepared for immunoprecipitation by washing 3× in M9 buffer, resuspension in 1 volume of TBS containing 1 mM PMSF and cOmplete protease inhibitor cocktail (Roche), and dropwise addition to liquid nitrogen to snap freeze. Frozen droplets were ground to a fine powder with a mortar and pestle and resuspended in an RIPA buffer with phosphatase inhibitors (50 mM Tris (pH 7.4), 150 mM NaCl, 1% NP-40, 0.1% SDS, 0.1% sodium deoxycholate, 10 mM EDTA, 1 mM EGTA, 1 mM Na$_3$VO$_4$, 60 mM β-glycerophosphate, 2.5 mM sodium pyrophosphate, 50 mM sodium fluoride, 2 mM benzamidine, 1 mM PMSF, and cOmplete protease inhibitor cocktail). Lysates were sonicated to complete worm lysis and cleared by centrifugation at 60k xg. GFP was pulled down with GFP-trap agarose beads (ChromoTek) and mScarlet was pulled down with RFP-trap agarose beads (ChromoTek). Select samples were treated on-bead with lambda protein phosphatase (NEB) in the manufacturer's PMP buffer to remove all phosphorylation. Immunoprecipitated samples were eluted in SDS sample buffer, run on 4% to 12% Bis-tris gels (Invitrogen), transferred to PVDF membranes, and blotted for GFP (Roche 11814460001), mScarlet (ChromoTek 6G6), or phospho-serine/threonine (Abcam ab17464).

Purified protein constructs were produced in Rosetta2(DE3) *E. coli* cells grown at 37˚C in TB medium and induced with 0.4 mM IPTG overnight at 18˚C. Cells were lysed in 20 mM Tris (pH 7.4), 500 mM NaCl (high salt to inhibit phase separation), 5 mM β-mercaptoethanol, 1 mM phenylmethylsulfonyl fluoride (PMSF), and cOmplete protease inhibitors (Roche) with 0.2 mg/ml lysozyme. Lysates were spun at 30,000 xg to clear and remove any pre-condensed or aggregated protein. His6-GFP and His6-mCherry proteins were purified with cOmplete His tag resin (Roche) according to the manufacturer's protocols as previously described [11]. pMTH proteins containing MBP on their N-terminus and 8xHis on their C-terminus were purified in a two-step process to enrich for full-length products. First, lysates were incubated with cOmplete His tag resin (Roche) for 2 h, washed 3× (20 mM Tris (pH 7.4), 500 mM NaCl, 5 mM β-mercaptoethanol), and eluted in wash buffer plus 250 mM imidazole. Eluted samples were subsequently bound to amylose resin (NEB) for 2 h, drained in a polyprep column (Biorad), washed with 12 volumes of wash buffer (20 mM Tris (pH 7.4), 500 mM NaCl, 5 mM β-mercaptoethanol), and eluted with wash buffer plus 10 mM maltose. Samples were dialyzed overnight with 10,000 MWCO SnakeSkin dialysis tubing (Thermo), concentrated to roughly 50 to 100 μm with 10,000 or 100,000 MWCO Amicon Ultra centrifugal filters (Millipore), and snap frozen in aliquots at −80˚C.

In vitro kinase assays (modified from [31]) consisted of 1 μg substrate, 1 μg SAD-1 kinase, 0.1 μg LKB-1 complex (Sigma-Aldrich), 500 μm ATP, and 2 μCi γ$^{32}$P-ATP (Perkin-Elmer) in 25 mM Tris (pH 7.4), 10 mM magnesium acetate, 1 mM DTT. SAD-1 and LKB-1 were preincubated to activate the kinase in the absence of γ$^{32}$P-ATP for 30 min at room temperature, followed by addition of the substrate and γ$^{32}$P-ATP for an additional 30 min at room temperature. Reactions were quenched with sample buffer and run on a 4% to 12% Bis-tris gel and stained with Coomassie SimplyBlue (Invitrogen) before drying. Dried gels were assembled in cassettes with a BAS-IP Phosphor screen (Cytiva) overnight and imaged on an Amersham Typhoon system (Cytiva) with 635 nm excitation.

In vitro liquid–liquid phase separation assays were performed as previously described [11]. Purified recombinant proteins were diluted to physiological salt conditions (20 mM Tris (pH 7.4), 150 mM NaCl) in the presence of 10% PEG 3000. Phase diagrams were determined on the basis of proteins' ability to form droplets within 5 min in each condition. Phase boundaries in the data were determined computationally with support vector machine learning using a third order polynomial kernel implemented in Python with Scikit-learn.

The in vitro binding assay was performed by pre-binding 1 μg of each purified recombinant GFP construct to GFP-trap agarose beads in 20 mM Tris (pH 7.4), 500 mM NaCl, 5 mM β-mercaptoethanol. Select GFP samples were pre-phosphorylated by the SAD-1 kinase as described above (excluding $\gamma^{32}$P-ATP), and 1 μg of each MBP-conjugated prey protein was incubated with the GFP-SAM-bound beads for 1 h. Beads were washed sparingly, eluted with SDS sample buffer, run on 4% to 12% Bis-tris gels (Invitrogen), and stained with Coomassie SimplyBlue (Invitrogen).

## Phosphoproteomics

*C. elegans* were prepared for phosphoproteomics as described previously [48]. $^{14}$N *sad-1Δ* and $^{15}$N-labeled wild-type worms were mixed at a 1:1 volume ratio. The sample mixtures were lysed by resuspension in 2× RIPA with 2× EDTA-free proteinase inhibitor cocktail (Roche) and 2× PhosSTOP (Roche). Samples were homogenized with a FastPrep-24 (MP Biomedicals) and cleared by centrifugation at 20,000 xg for 30 min. Proteins were precipitated from cleared samples with acetone and resuspended in a urea solution (8 M urea, 100 mM Tris-HCL (pH 8.5)) for trypsin digestion.

Peptides were prepared for mass spectrometry as previously described [48]. Briefly, 10 mg of protein ($^{14}$N and $^{15}$N mixtures) was reduced with 5 mM TCEP, alkylated with 10 mM iodoacetamide, and digested with trypsin overnight at 37°C. Digested peptides were separated into 12 fractions on a Xtimate C18 reverse phase HPLC column (10 × 250 mm, 5 μm, Welch Materials) with an Agilent 1200 Series HPLC. Each fraction was acidified and enriched for phosphopeptides using a PolyMAC-Ti Enrichment Kit (Tymora Analytical). The resulting phosphopeptides were resolved in 0.25% formic acid buffer for mass spectrometry analysis.

Each fraction was analyzed in 2 technical replicates by a Q-Exactive mass spectrometer (Thermo Fisher Scientific) interfaced with an Easy-nLC1000 reverse phase chromatography system (Thermo Fisher Scientific). Data acquisition, phosphopeptide identification, and quantification were performed exactly as previously described [48]. A wild-type dataset from this former study ($^{14}$N wild type mixed with $^{15}$N wild type sample) was used to normalize for altered phosphoproteomes in $^{15}$N samples. The dataset has been deposited to the ProteomeXchange via iProX (PXD043081). GO enrichment analysis was performed with WormCat [57].

In vitro kinase assay phosphosite mapping was performed by MS Bioworks (Ann Arbor, Michigan, United States of America). Phosphorylated SYD-2 constructs in gel slices were reduced with 10 mM dithiothreitol (DTT), alkylated with 50 mM iodoacetamide, and digested with either trypsin, chymotrypsin, or elastase (Promega). Samples were analyzed by nano LC-MS/MS on a Waters M-Class HPLC coupled to a Thermo Fisher Orbitrap Fusion Lumos mass spectrometer. Peptides were loaded on a trapping column and eluted over a 75 μm analytical column at 350 nL/min. Both columns were packed with Luna C18 resin (Phenomenex). The mass spectrometer was operated in data-dependent mode, with the Orbitrap operating at 60,000 FWHM and 15,000 FWHM for MS and MS/MS, respectively. APD was enabled and the instrument was run with a 3 s cycle for MS and MS/MS. Dataset is available at MassIVE (MSV000091471).

## Microscopy

Fluorescence imaging was performed on a Zeiss LSM 980 Airyscan 2 system with a 63× Plan-Apochromat 1.4NA objective and 405 nm, 488 nm, or 561 nm lasers. The Airyscan 4Y multiplex mode was used to capture superresolution images of HSN synapses, while Airyscan 8CO confocal mode was used for embryonic synapse FRAP to maximize sensitivity. In vitro condensate FRAP was performed with the confocal LSM detector. Animals were imaged live on 5% agarose pads in 1 mM levamisole.

## Image analysis

Airyscan images were processed with Zeiss ZEN software. Image analysis was performed in ImageJ/Fiji (National Institutes of Health). FRAP curves were calculated after background and time-course photobleaching correction and normalized to 100% for visualization. HSN intensity quantification was performed on background-subtracted sum projections of Z-stacks. Cell body puncta quantification was performed with Fiji's 3D Object Counter plugin. Representative images are maximum intensity projections that have been cropped and rotated where necessary. Images have inverted grayscale or "fire" Fiji lookup tables applied for visualization.

## Statistics and reproducibility

Statistical comparisons were performed with one-way ANOVA tests and multiplicity-corrected $P$-values were calculated versus wild type using Dunnett's test in Prism 9 (GraphPad). Bar graphs depict all data points, means, and 95% confidence interval error bars. FRAP graphs depict 95% confidence intervals and one-phase association least-squares fits performed in GraphPad Prism 9. Each $C. elegans$ in vivo measurement (FRAP or fluorescence intensity) is a biological replicate taken from a distinct animal. All in vivo imaging data were replicated in at least 2 independent sessions. All in vitro experiments were replicated at least twice. Three biological replicates were performed for SAD-1 phosphoproteomics. Raw data underlying all graphs is available in S1 Data. Uncropped blot images are available in S1 Raw Images.

## Supporting information

**S1 Fig. Development of synapses in the *C. elegans* hermaphrodite-specific neuron.** (A) Representative Airyscan superresolution images of the indicated endogenously tagged synaptic proteins at 3 developmental time points. Brightfield images show vulval morphology used to identify each time point. Hours indicate time elapsed since starved and synchronized L1 animals were reintroduced to food at 20˚C. (B) Quantification of HSN synapse intensity of each marker from (A). Underlying data is available in S1 Data.
(TIF)

**S2 Fig. Catalytically inactive lambda phosphatase control.** (A) Fluorescence recovery after photobleaching of endogenous mScarlet-SYD-2-λpptase$^{H76N}$ catalytically inactive control at embryonic nerve ring synapses to measure dynamics. Scale bars, 5 μm. (B) Quantification of FRAP in (A). Wild-type and λpptase data included from Fig 1C. Underlying data is available in S1 Data.
(TIF)

**S3 Fig. SAD-1 kinase regulation of synaptic vesicle clustering.** Synapse formation phenotypes visualized with Airyscan superresolution imaging of endogenous GFP-RAB-3 in the indicated mutants. Scale bars, 5 μm.
(TIF)

**S4 Fig. SAD-1 phosphorylation of SYD-2's IDR in vitro is dispensable for in vivo function.** (A) SYD-2 phosphosites identified from in vivo phosphoproteomics (yellow) or in vitro kinase assays (burgundy). Sites were grouped based on location for subsequent testing. See S1 and S2 Tables. (B) In vitro kinase assay between SAD-1 and SYD-2 IDR with or without phosphosite mutations. SAD-1 is activated by the LKB-1 kinase complex. (C) HSN synapse formation phenotypes visualized with Airyscan superresolution imaging of endogenous fluorescent tags in the indicated mutants. Scale bars, 5 μm. (D) Quantification of HSN intensities in (C). No significant difference in synapse formation was seen in IDR phosphomutants. (E) Aldicarb

synaptic transmission assay shows no defects in an SYD-2 IDR phosphomutant. Extended time to paralysis on 1 mM Aldicarb indicates defective synaptic transmission. $n > 20$ for each genotype. Underlying data is available in S1 Data.
(TIF)

**S5 Fig. Additional SYD-2 phosphorylation analysis.** (A) HSN synapse formation phenotypes visualized with Airyscan superresolution imaging of endogenous GFP-RAB-3 in the indicated mutants. Scale bars, 5 μm. Quantification is presented in Fig 4B. (B) Alignment of *C. elegans*, *D. melanogaster*, and *H. sapiens* SYD-2/Liprin-αs showing conservation of 3 SAM phospho-sites.
(TIF)

**S6 Fig. Constitutive activation of SYD-2 leads to loss of synaptic localization.** (A) HSN synapse formation phenotypes visualized with Airyscan superresolution imaging of endogenous fluorescent tags in the indicated mutants. Removal of SAM domains or mimicking SAD-1 phosphorylation leads to reduced SYD-2 at HSN synapses. See quantification in Fig 6B. Scale bar, 5 μm. (B) Aldicarb synaptic transmission assay. Extended time to paralysis on 1 mM Aldicarb indicates defective synaptic transmission. $n > 20$ for each genotype. (C) HSN synapse formation phenotypes visualized with Airyscan superresolution imaging of endogenous fluorescent tags in the indicated mutants; 10 μm of the proteasome inhibitor bortezomib or DMSO as a control was added to the indicated animals for 4 h prior to imaging. Scale bar, 5 μm. (D) Quantification of SYD-2 synaptic intensities in (C). Underlying data is available in S1 Data.
(TIF)

**S1 Data. Underlying data for Figs 1–8 and S1–S6.**
(XLSX)

**S1 Table. SAD-1 kinase in vivo phosphoproteomics.**
(XLSX)

**S2 Table. SAD-1 in vitro kinase assay phosphoproteomics.**
(XLSX)

**S3 Table. *C. elegans* strains used in this study.**
(XLSX)

**S4 Table. DNA constructs used in this study.**
(XLSX)

**S1 Raw Images. Raw gel images for Figs 1, 3, 7 and S4.**
(PDF)

## Acknowledgments

We thank Wen-Jun Li for assistance with mass spectrometry data and Mardo Koivomagi and Jan Skotheim for assistance with radiolabeled kinase assays.

## Author Contributions

**Conceptualization:** Nathan A. McDonald, Kang Shen.

**Formal analysis:** Nathan A. McDonald, Li Tao.

**Funding acquisition:** Nathan A. McDonald, Kang Shen.

**Investigation:** Nathan A. McDonald, Li Tao.

**Methodology:** Nathan A. McDonald.

**Project administration:** Nathan A. McDonald.

**Supervision:** Meng-Qiu Dong, Kang Shen.

**Writing – original draft:** Nathan A. McDonald.

**Writing – review & editing:** Nathan A. McDonald, Li Tao, Meng-Qiu Dong, Kang Shen.

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
