## [Editor Report · Decision Letter 0]

10 Jul 2023

Dear Dr Shen, 

Thank you for submitting your manuscript entitled "SAD-1 kinase controls presynaptic phase separation by relieving SYD-2/Liprin-α autoinhibition" for consideration as a Research Article by PLOS Biology. Please accept my apologies for the delay in getting back to you as we consulted with an academic editor about your submission.

Your manuscript has now been evaluated by the PLOS Biology editorial staff, as well as by an academic editor with relevant expertise, and I am writing to let you know that we would like to send your submission out for external peer review.

Once your full submission is complete, your paper will undergo a series of checks in preparation for peer review. After your manuscript has passed the checks it will be sent out for review. To provide the metadata for your submission, please Login to Editorial Manager (https://www.editorialmanager.com/pbiology) within two working days, i.e. by Jul 12 2023 11:59PM.

Kind regards,

Richard

Richard Hodge, PhD

rhodge@plos.org

PLOS

---

## [Decision Letter · Decision Letter 1]

9 Aug 2023

Dear Dr Shen,

Thank you for your patience while your manuscript "SAD-1 kinase controls presynaptic phase separation by relieving SYD-2/Liprin-α autoinhibition" was peer-reviewed at PLOS Biology. Please accept my apologies for the delays that you have experienced during the peer review process. Your manuscript has now been evaluated by the PLOS Biology editors, an Academic Editor with relevant expertise, and by three independent reviewers. 

In light of the reviews, which you will find at the end of this email, we would like to invite you to revise the work to thoroughly address the reviewers' reports.

As you will see, the reviewers are generally very positive about the study and note that the findings are important and convincing. Whilst Reviewer’s #1 and #3 list several minor comments requesting additional discussions and reporting details, Reviewer #2 raises concerns with the overall strength of the phase separation aspects of the study and the quality of the proteins used in the biochemical analysis. After discussions with the Academic Editor, we ask that you address the comments from Reviewer #2 regarding the degradation of the product with new experiments or by caveating the limitations/interpretation of the findings in the manuscript text if this is the highest quality product that is accessible with the current experimental design. In addition, we will not make an exploration of whether mammalian liprin proteins also undergo phase separation essential to consider the revised version. 

Given the extent of revision needed, we cannot make a decision about publication until we have seen the revised manuscript and your response to the reviewers' comments. Your revised manuscript is likely to be sent for further evaluation by all or a subset of the reviewers.

**IMPORTANT - SUBMITTING YOUR REVISION**

*Re-submission Checklist*

*Published Peer Review*

*PLOS Data Policy*

*Blot and Gel Data Policy*

Sincerely,

Richard

Richard Hodge, PhD

rhodge@plos.org

REVIEWS:

Reviewer #1 (Stephan Sigrist, signs review): Every presynapse builds a core active zone scaffold, where ion channels cluster and synaptic vesicles release their neurotransmitters. The protein composition of active zones (AZs) is rather well characterized, with Syd-2 aka Liprin, ELKS/BRP and RIM-BP proteins being conserved core scaffold components. Recent, prominently published work by the Shen lab has shown that Syd-2/ELKS "scaffold" adopts a liquid character in association with driving developmental AZ assembly, but later matures into a more solid structure. Regulatory principles confining such liquid-solid phase transitions to properly steer active zone assembly so far remain largely enigmatic, however. 

Here, using a broad spectrum of genetic and biochemical methods the Shen lab investigates the role of the Sad-1 kinase in the control of Syd-2 liquidity and thus AZ scaffold assembly. 

They in my eyes provide convincing evidence for Sad-1 phosphorylation of Syd-2 at/within the first Syd-2 SAM-domain controls AZ assembly in conjunction with controlling liquidity. 

This study in my eyes provides important evidence to exemplify Syd-2 phosphorylation dependent recruitment of "downstream components" such as Rim/Unc10 and Rab3. They convincingly show that SYD-2's SAM domains directly interact with its N-terminus to inhibit phase separation, which is released by specific SAD-1 phosphorylation in the first SAM domain.

This becomes possible by a real tour de force using an impressive array of on-locus point mutations in syd-2. I do recommend publication of this study in Plos Biology essentially as is. 

Comments: 

As said, I am very positive and suggest publication. This said, the object of study, though timely and relevant, is intrinsically difficult and drawing undisputably causal relations still challenging (though they are getting close by doing domain swops). In my eyes, extrapolating local viscosity/liquidity within AZs by "global" FRAP is a somewhat coarse approach, which might mask different AZ subzones and subpopulations of the relevant proteins. This reviewer would appreciate to see this somewhat more reflected in the discussion of their results. 

Additional points: 

- Do the in vitro and in vivo identified Sad-1 sites are in line with a certain consensus? Following this, are the putatively artefactual IDR phospho-sites less close to a consensus? Concerning their homology considerations, I do suggest to also mention the Drosophila situation concerning the relevant Sad-1 sites, as Drosophila also is a highly relevant model in AZ research. 

- They say: "Wildtype SYD-2 at nascent synapses recovers quickly after photobleaching, due to liquid condensate formation and rapid exchange between condensate and cytoplasmic pools". How do they know that there is exchange with cytoplasmic pools? Couldn't it be all exchange between AZs? I am also rather confident that extra-AZ pools do exist, but would appreciate to hear their arguments in this point. 

- What exactly are the arguments that it is the IDR entertaining dominating critical interactions to other active zone proteins? I thought the more N-terminal coiled coil domains of Syd-2/Liprin might be relevant (as well)?

- Again concerning discussion: do they think of the AZ liquidity to be reversible? Interestingly, the Nter + SAM droplets formed at high concentration were not dynamic but resembled aggregates rather than liquid condensates. Thus: is the transport of Syd-2 really operating in a closed form, not associated with other AZ proteins?

- Which states do the differentiate: liquid condensate to recruit, "aggregate" state once recruitment is ended and AZs are matured? Do AZ scaffolds in Delta-SAM mutants loose liquidity prematurely? Are there extra-AZ axonal/cell body aggregates forming for any of their mutants? Especially the phosphomimetic mutants which should suffer "premature liquidity"?

- How do they think concerning ELKS, major subject of investigation in their previous LLPS study: transporting independent of Syd-2? Also in an aggregated form? SRPK might keep ELKS/BRP in a liquid transport form in Drosophila, do they think ELKS/RimBP complexes might transport differently from Syd-2? 

- "A complete 10A mutant reproduced the synapse formation defects of the 3A mutant" it is somewhat hard to keep track with all their genotype nomenclature, maybe a little more of in between explanation/reiteration might be helpful here. 

Reviewer #2: In this manuscript, McDonald et al present their interesting and intriguing findings on the role SAD-1 kinase in regulating SYD-2 and presynaptic assembly in C. elegans. They used a combination of genetic, biochemical, and imaging approaches to show that SAD-1 phosphorylates SYD-2 on three serine residues in the first SAM domain, which relieves an autoinhibitory interaction between the N-terminal coiled coils and the C-terminal SAM domains. The authors also showed that SAD-1 is localized to nascent synapses by upstream synaptic cell adhesion molecules. Their data provides evidence for the novel regulation of SYD-2 activation by a presynaptic kinase. Considering the high conservation of the three phosphorylation sites in the liprin-alpha family, it is plausible that the phosphorylation-mediated activity regulation may extend to mammalian liprin-alpha proteins. Thus, this study is important. However, the proposed phase separation-based mechanism lacks robust support from their data, and an alternative explanation involving the regulation of specific protein-protein interactions could also be considered. I recommend the authors to strengthen the phase separation part of the manuscript and provide additional evidence to support their claim.

Major concerns

1. The authors propose that SYD-2 phase separation is regulated by Nter-SAM interaction. However, the related data presented in Figure 6 is not convincing. The main concern lies in the low quality of the proteins used in the biochemical analysis, which raises doubts about the reliability of the results.

a. The Nter protein they purified exhibited severe degradation, appearing as several bands in SDS-PAGE (Fig. 6E). The target protein was not even the major band observed. The use of such a degraded sample for the phase separation analysis may introduce artificial outcome or biased results. It is necessary for the authors to obtain a high-quality sample of Nter, free from degradation, to ensure the accuracy and reliability of their findings.

b. The GFP-SAM protein used in the biochemical assays was also affected by partial degradation as shown in Fig. 6E. It further compromises the reliability of their interpretation of both the phase separation and pulldown results.

c. Again, in Fig. 6E, it is odd to see that all the degraded fragments of Nter were pulled down by the SAM domains. Could it be that all these degraded fragments contain SAM-binding sequence? Or, more likely, such a heavily degraded mixture nonspecifically associated with beads. The use of high-quality samples will be crucial to address this issue.

d. In the presence of the SAM domains, they observed a decrease in the dynamics of Nter, as indicated by FRAP analysis (Fig. 6D). The uneven distribution of Nter protein in the condensate (Fig. 6B) also supports the alteration in protein dynamics. However, it is essential to note that a decrease in the dynamics of protein condensates does not necessarily imply a reduced ability to phase separate. 

e. The authors state that "binding between domains surrounding the IDR prevents its phase separation" and suggest that the release of the Nter-SAM interaction could lead to an alteration in the IDR conformation, promoting phase separation. For such an alteration to occur, it would require the surrounding domains of IDR to be present on both sides. However, the IDR was not covalently linked with the SAM domains in the experiment shown in Fig. 6B-D that was used to support the importance of Nter-SAM interaction for the inhibition of IDR-mediated phase separation. Also, considering SYD-2 exists as an oligomer mediated by the coiled coils, the Nter-SAM interaction may occur intermolecularly. In this context, the interaction could promote rather than decrease its phase separation by providing intermolecular binding valence.

2. The data in Fig. 6F&G is interesting but also provocative. The authors showed that the IDR of SYD-2 can be functionally replaced by FUS IDR. It can then be deduced that the SYD-2 SAM domains should also bind to FUS IDR, a highly unusual predication that needs to be verified by experiments. 

3. The three important serine residues are located at the SAM1 domain, a well-characterized region known for protein-protein interactions. It raises the intriguing possibility that phosphorylation of these sites may have an impact, either positive or negative, on these interactions. Among these sites, the S896 residue is close to the liprin-beta binding surface, while S904 is located near the CASK-binding surface. The synaptic defects observed in the SAD1 deletion may be attributed to the dysregulation of SAM-mediated interactions. In the article, the authors should acknowledge such possibilities in the Discussion. 

Minor points

1. To comprehensively present the potential effects of phosphorylation, it would be beneficial to indicate the previously characterized binding surfaces in Fig. 3C.

2. Could the author add the phosphomimetic mutant SAM(3E) into the Nter condensate? This experiment will further validate whether phosphorylation regulates the interaction between SAM and Nter.

3. Exploring whether mammalian liprin-alpha proteins undergo phase separation triggered by the phosphorylation of the three serines would be valuable to establish the broader relevance of their findings. To test the phosphorylation-dependent phase separation, the authors could consider overexpressing liprin-alpha and its phosphomimetic mutant in mammalian cells.

Reviewer #3: In a landmark paper published in 2020, these authors reported that scaffold proteins at the active zone undergo liquid-liquid phase separation during synapse development in vivo, a process that is further critical for incorporating additional components into the active zone. Here, they provide compelling evidence that SAD-1, a kinase known to regulate synapse formation in different organisms, phosphorylates the active zone protein SYD-2/liprin. This phosphorylation disrupts an intramolecular interaction within SYD-2, triggering LLPS. Based on their experimental data, they propose that SAD-1 is positioned upstream of the active zone formation process and is itself located at nascent synapses directly or indirectly through adhesion molecules. This model is based on convincing data, and the work is of high technical quality, combining biochemistry, genome engineering and high -resolution microscopy in vivo, which is likely unachievable in any other organism. 

I only have few comments:

- I am puzzled by the extreme variability in fluorescence values throughout this study. For instance, in Fig 2G, the standard deviation of mScar-SYD-2 % intensity values under the control condition is 48, with extreme values from 9 to 229. What could be the biological significance of such fluctuations? It is mentioned in the Materials and Methods section that the images are derived from "synchronized early L4" animals. Could this possibly correspond to an early phase of synaptic maturation, which might explain why even minor developmental shifts could have a strong impact? Providing a kinetics analysis of the accumulation of the different markers (SYD-2, UNC-10, RAB-3), at least in the wild type, would provide insights to rule out the possibility that various genetic conditions and mutations merely postpone or expedite synapse formation.

- the dependence of SAD-1 on SYG-1, but not SYD-2, for its synaptic localization is an important result. Yet, as mentioned by the authors, these results are inconsistent with those previously published in Patel 2006. The explanation provided attributes this disparity to the prior utilization of overexpressed reporters. It would be interesting to analyze the distribution of SAD-1 in syd-1 mutants to test if SAD-1 remains dependent of SYD-1 for its localization, or if this was also an artefact.

- from the initial SYG-1 paper (2003), SYG-1 concentration peaked at presumptive synaptic sites several hours before the detection of synaptic vesicles. Again, it would be important to observe the kinetics of SYG-1 and SAD-1 localization during HSN synapse formation. Along these lines, do the authors speculate whether SAD-1 is constitutively active or if it requires local activation?

---

## [Editor Report · Decision Letter 2]

26 Oct 2023

Dear Dr Shen,

Thank you for your patience while we considered your revised manuscript "SAD-1 kinase controls presynaptic phase separation by relieving SYD-2/Liprin-α autoinhibition" for publication as a Research Article at PLOS Biology. This revised version of your manuscript has been evaluated by the PLOS Biology editors and the Academic Editor.

Based on our Academic Editor's assessment of your revision, I am pleased to say that we are likely to accept this manuscript for publication, provided you satisfactorily address the following data and other policy-related requests that I have provided below (A-D). In addition, we note that some of the valuable discussions provided in the rebuttal document that address remaining caveats and limitations of the work have not been included in the main manuscript. We ask that these arguments are added to the discussion section during this round of revision. 

(A) You may be aware of the PLOS Data Policy, which requires that all data be made available without restriction: http://journals.plos.org/plosbiology/s/data-availability. For more information, please also see this editorial: http://dx.doi.org/10.1371/journal.pbio.1001797

-Supplementary files (e.g., excel). Please ensure that all data files are uploaded as 'Supporting Information' and are invariably referred to (in the manuscript, figure legends, and the Description field when uploading your files) using the following format verbatim: S1 Data, S2 Data, etc. Multiple panels of a single or even several figures can be included as multiple sheets in one excel file that is saved using exactly the following convention: S1_Data.xlsx (using an underscore).

-Deposition in a publicly available repository. Please also provide the accession code or a reviewer link so that we may view your data before publication. 

Figure 1D, 1G, 2B, 2D, 4B-C, 4E, 5B, 6B, 6D-E, 7C-D, 7G, 8B, S1B, S2B, S4D-E, S6B, S6D

(B) Please also ensure that each of the relevant figure legends in your manuscript include information on *WHERE THE UNDERLYING DATA CAN BE FOUND*, and ensure your supplemental data file/s has a legend.

(C) Please ensure that your Data Statement in the submission system accurately describes where your data can be found and is in final format, as it will be published as written there. 

(D) Please note that per journal policy, the model system/species studied should be clearly stated in the abstract of your manuscript.

We expect to receive your revised manuscript within two weeks. 

*Published Peer Review History*

*Press*

Sincerely,

Richard

Richard Hodge, PhD

rhodge@plos.org

PLOS

---

## [Editor Report · Decision Letter 3]

6 Nov 2023

Dear Dr Shen,

On behalf of my colleagues and the Academic Editor, Cody Smith, I am pleased to say that we can accept your manuscript for publication, provided you address any remaining formatting and reporting issues. These will be detailed in an email you should receive within 2-3 business days from our colleagues in the journal operations team; no action is required from you until then. Please note that we will not be able to formally accept your manuscript and schedule it for publication until you have completed any requested changes.

PRESS

Best wishes, 

Richard

Richard Hodge, PhD

rhodge@plos.org

PLOS
